# Development of an agar-plug cultivation system for bioactivity assays of actinomycete strain collections

Nico Ortlieb[1,2,3¤a], Elke Klenk[1¤b], Andreas Kulik[1,4], Timo Horst Johannes Niedermeyer[1,2,3]*

**1** Department of Microbiology/Biotechnology, Interfaculty Institute of Microbiology and Infection Medicine, Eberhard Karls University Tübingen, Tübingen, Germany, **2** German Centre for Infection Research (DZIF), Partner Site Tübingen, Tübingen, Germany, **3** Department of Pharmaceutical Biology/Pharmacognosy, Institute of Pharmacy, Martin-Luther-University Halle-Wittenberg, Halle (Saale), Halle, Germany, **4** Department of Microbial Bioactive Compounds, Interfaculty Institute of Microbiology and Infection Medicine, Eberhard Karls University Tübingen, Tübingen, Germany

¤a Current address: Bayer AG, Product Supply – Pharmaceuticals, Pharmaceutical Affairs, Berlin, Germany
¤b Current address: Department of Microbial Interactions, IMIT/ZMBP, Eberhard Karls University Tübingen, Tübingen, Germany
* timo.niedermeyer@pharmazie.uni-halle.de

**Data Availability Statement:** All relevant data are within the manuscript and its Supporting information files.

## Abstract

Natural products are an important source of lead compounds for the development of drug substances. Actinomycetes have been valuable especially for the discovery of antibiotics. Increasing occurrence of antibiotic resistance among bacterial pathogens has revived the interest in actinomycete natural product research. Actinobacteria produce a different set of natural products when cultivated on solid growth media compared with submersed culture. Bioactivity assays involving solid media (e.g. agar-plug assays) require manual manipulation of the strains and agar plugs. This is less convenient for the screening of larger strain collections of several hundred or thousand strains. Thus, the aim of this study was to develop a 96-well microplate-based system suitable for the screening of actinomycete strain collections in agar-plug assays. We developed a medium-throughput cultivation and agar-plug assay workflow that allows the convenient inoculation of solid agar plugs with actinomycete spore suspensions from a strain collection, and the transfer of the agar plugs to petri dishes to conduct agar-plug bioactivity assays. The development steps as well as the challenges that were overcome during the development (e.g. system sterility, handling of the agar plugs) are described. We present the results from one exemplary screening campaign targeted to identify compounds inhibiting Agr-based quorum sensing where the workflow was used successfully. We present a novel and convenient workflow to combine agar diffusion assays with microtiter-plate-based cultivation systems in which strains can grow on a solid surface. This workflow facilitates and speeds up the initial medium throughput screening of natural product-producing actinomycete strain collections against monitor strains in agar-plug assays.

**Funding:** NO and THJN were funded by the German Center for Infection Research (DZIF; TTU09.706). We acknowledge financial support within the funding programme Open Access Publishing by the German Research Foundation (DFG). The funders had no role in study design, data collection and analysis, decision to publish, or preparation of the manuscript.

**Competing interests:** The authors have declared that no competing interests exist.

## Introduction

Natural products are an indispensable source of lead compounds for the development of drug substances [1, 2]. About 60% of the known natural products with biological activity are derived from plants, about 40% are produced by microorganisms such as fungi and bacteria [3]. Of the bacterial natural product producers, especially in the field of antibiotics discovery, the most famous are undoubtedly the Gram-positive, filamentous actinomycetes [3–6], although other bacteria such as myxobacteria and cyanobacteria have received increasing attention over the last years [7–9]. Both major pharmaceutical companies and academic groups have created impressive actinomycete strain collections, sometimes comprising of tens of thousands of strains [10–12]. Extracts of these strains have been screened against many targets over the years. One famous academic strain collection is the collection at the University of Tübingen, established by Prof. Hans Zähner [13–15]. Important compounds such as deferoxamine [16], a compound used to treat iron intoxication that is listed in the WHO Model List of Essential Medicines [17], or the herbicide glufosinate [18] resulted from research on this strain collection.

However, the efficacy of screening campaigns using these actinomycete strain collections, especially screenings for novel antibiotics, has dropped in the last decades, and most major pharmaceutical companies have discontinued their antibiotics discovery programs [19–21]. It has become more and more difficult to isolate novel compounds from actinomycetes, high rates of rediscovery of already known compounds being one of the major issues [22]. On the other hand, resistances of human pathogens against the medically used antibiotics are currently on the rise e.g. due to misuse of antibiotics in animal farming, increased mobility of people in a globalized world, and increased industrialization [23], resulting in the current situation of an "antibiotic crisis" and the fear of return to the pre-antibiotic era [24, 25]. This makes the search for novel antibiotic natural products mandatory. In addition to compounds with direct antibacterial activity, inhibitors of bacterial quorum sensing (QS) are currently discussed as alternative anti-infectives [26, 27]. As QS-inhibitors do not possess direct antibacterial activity, they are less likely to induce resistance development (lower selection pressure), but at the same time reduce virulence and biofilm formation. Thereby, QS-inhibitors enable the host's immune system to clear the pathogen effectively while at the same time they have little to no impact on the native microbiota.

In the past, mainly extract, fraction, or pure natural product libraries have been used in bioactivity screenings [28, 29]. It is known that most actinomycetes can produce a large diversity of natural products, and individual strains can produce many different compounds depending on growth conditions (e.g. medium composition, culture morphology, cultivation time, stressors) [3, 30–32]. The cultivation conditions direct the natural product profiles of the resulting extracts. Thus, uncommon cultivation conditions can increase the chances of producing novel chemistry [33, 34]. Extracts in the past have most often been generated by extraction of the cultivation medium of submersed actinomycetes cultures with organic solvents such as ethyl acetate. Although it is well known that Actinobacteria produce a different set of natural products when cultivated on solid growth media (agar) compared with submersed culture [35], this has less frequently been used to generate extracts for bioactivity screening, because extracting solid media of a large number of strains is inconvenient. One convenient way to circumvent extraction of the agar prior to bioactivity testing is to study the antibiotic activity of actinomycete natural products secreted into a solid growth medium in an agar plug assay [36]: A natural product producing actinomycete strain is grown on agar. After a defined incubation time, a plug of the agar including the strain is punched out aseptically using e.g. a cork borer, the back of a pipet tip, or a trimmed syringe. The agar plug is transferred to a fresh petri dish, and

embedded into agar containing a monitor bacterium. In the case of the production of e.g. an antibacterial natural product, the monitor strain is not growing in the agar around the agar plug with the actinomycete, because antibacterial compounds secreted into the agar during actinomycete growth diffuse into the surrounding fresh agar. However, as this assay requires manual manipulation of the strains and agar plugs, it is not feasible to be used with a larger strain collection of several hundred strains at once.

Small-scale cultivation systems using microtiter plates (MTPs) are widely used in microbiology laboratories [37, 38]. The Duetz cryo-replicator is a system for the parallel handling of 96 bacterial strains in MTPs [39]. Its main use is to inoculate liquid medium in MTPs with cryo-preserved strains in strain collection master-plates for subsequent studies [40, 41]. Although this and similar systems have been used with *Streptomyces* sp. strains before [42–44], these cultivation systems have been used only for submersed cultivation, and have never been used to transfer actinomycetes spore suspensions to solid agar for subsequent agar plug assay screenings.

The aim of the study presented here was to remove the main bottleneck of agar plug assays, the inconvenient manual agar plug handling. We describe the successful development of a medium-throughput cultivation and agar-plug assay workflow that allows the convenient inoculation of solid agar plugs with actinomycete spore suspensions from a strain collection, and the transfer of the agar plugs to petri dishes to conduct agar-plug bioactivity assays.

## Materials and methods

### Preparation of spore suspension master plates

Actinomycetes were grown in petri dishes (Ø 94 mm) on ISP2 agar until they showed strong sporulation. The spores were collected by addition of 5 mL sterile 37.5% glycerol in water (v/v). 1.5 mL of the resulting spore suspensions were transferred into MTPs (polystyrene transparent square 96-half-deepwell microplates, flat bottom; EnzyScreen). The MTPs were closed with removable silicone mats (CapMat, Greiner Bio-One) and stored at -80°C.

**Media compositions.** ISP media according to the International Streptomyces Project were used to cultivate the actinomycete strains [45]. All media were prepared by dissolving the following ingredients in 1 L of distilled water. ISP2: 4 g/L yeast extract (Ohly GmbH, Germany), 4 g/L dextrose (Carl Roth, Germany), 10 g/L malt extract (Thermo Fischer Scientific, USA), pH 7.3. ISP3: 20 g/L oatmeal (Holo Hafergold, Reform Kontor GmbH & Co. KG, Germany), 5 mL trace metal standard solution (CaCl$_2$ · 2 H$_2$O 3 g/L, Fe(III)-citrate 1 g/L, MnSO$_4$ · H$_2$O 200 mg/L, ZnCl$_2$ 100 mg/L, CuSO$_4$ · 5 H$_2$O 25 mg/L, Na$_2$B$_4$O$_7$ · 10 H$_2$O 20 mg/L, Na$_2$MoO$_4$ · 2 H$_2$O 10 mg/L, CoCl$_2$ · 6 H$_2$O, 4 mg/L), pH 7.3. NL19: 20 g/L mannitol, 20 g/L soy meal (Henselwerk, Magstadt, Germany), pH 7.5. NL200: 20 g/L mannitol, 20 g/L corn steep powder (Marcor Development Corp., Carlstadt, USA), pH 7.5. NL300: 20 g/L mannitol, 20 g/L cotton seed powder (PharmaMedia Dr. Müller GmbH, Leimen, Germany), pH 7.5. NL333: 5 g/L dextrose, 10 g/L soluble starch (Carl Roth G,bH & Co. KG), 10 g/L malt extract, 3 g/L yeast extract, 3 g/L BactoPeptone (Becton Dickinson, Sparks, USA), 3 g/L NH$_4$NO$_3$, 2 g/L CaCO$_3$, pH 7.0. NL400: 10 g/L dextrose, 2 g/L starch (Carl Roth G,bH & Co. KG), 3 g/L Bacto-Peptone, 5 g/L yeast extract, 3 g/L meat extract, 3 g/L CaCO$_3$, pH 7.0. NL410: 10 g/L dextrose, 10 g/L glycerol, 5 g/L oatmeal, 10 g/L soy meal, 5 g/L yeast extract, 5 g/L BactoCasaminoacid (Becton Dickinson), 1 g/L CaCO$_3$, pH 7.0. NL500: 10 g/L soluble starch, 10 g/L dextrose, 10 g/L glycerol, 15 g/L fish meal (Becton Dickinson), 10 g/L sea salt (Sigma-Aldrich), pH 8.0. 2% to 2.5% of agar (Sigma-Aldrich, USA) were added for solid media. TSB (tryptone soy broth; Termo Fisher Scientific), TSA (tryptone soy agar; Becton Dickinson). All media were autoclaved at 121°C before use.

**Screening against *Staphylococcus aureus* PC322.** Bioactivity screening against *S. aureus* PC322 was performed according to Nielsen *et al.* [46] using the developed screening system. A *S. aureus* PC322 culture was prepared as follows: A single colony of *S. aureus* PC322, grown on TSA, was used to inoculate a pre-culture in 10 mL of TSB supplemented with 5 μg/mL erythromycin in a flask with baffles and shaken at 180 rpm at 37˚C for about 12 h. The pre-culture was used to inoculate the main culture in 10 mL TSB (1% v/v), which was kept shaking at 180 rpm at 37˚C for 16 h. This main culture was diluted to an $OD_{600}$ of 0.1. 2 mL of this diluted culture were added to a solution of 320 μL 5-bromo-4-chloro-3-indoxyl-β-D-galactopyranoside (X- Gal; 20 mg/ml in DMSO) and 4 μL erythromycin solution (50 mg/mL in ethanol) in 40 mL TSA. Subsequently, the mixture was poured into petri dishes, either around the agar plugs in 120 x 120 mm square petri dishes, or in round petri dishes (Ø 92 mm). In the latter case, after solidification, round wells (Ø 7 mm) were made using the back end of a 200 μL pipette tip and filled with 50 μl of sample solution. 50 μl hamamelitannin solution (2 mg/mL in DMSO) were used as positive control [47].

## Taxonomy of strain Tü2700

To obtain genomic DNA for sequencing, Tü2700 biomass was harvested from a main culture after 4 d of incubation. To isolate the genomic DNA, the Qiagen Genomic-tip 100/G kit (Qiagen, Germany) was used, following the manufacturer's protocol. 50 μl deionized $H_20$ were used for elution. The genome has been sequenced using a PacBio RSII sequencer (macrogen Inc.; reagents: SMRT cell 8Pac V3 and DNA Polymerase Binding Kit P6). The genome sequence was analyzed using autoMLST to generate a phylogeny tree [48].

## Cultivation of *Streptomyces* sp. Tü2700

Pre-cultures were prepared by inoculation of spores in NL410 in Erlenmeyer flasks (max. 20% media filling) equipped with baffle and spiral for three days at 180 rpm at 27˚C. Small-scale media screening cultures (10 mL) were inoculated with 1% (v/v) of pre-culture, and incubated in 100 mL Erlenmeyer flasks with baffle and spiral at 180 rpm at 29˚C. Production cultures were inoculated with 500 mL of pre-culture in 9.5 L of NL19 medium. Cultivation was done in a 10 L bioreactor (B20, B. Braun, Melsungen, Germany) under continuous stirring at a rotor speed of 200 rpm, and an airflow of 5 L/min at 27˚C. 4 mL of Ucolub N115® were added during the 7 days of incubation to reduce foam development. Every 24 h, samples were drawn and tested for bioactivity.

## Extraction and compound isolation

After harvesting, the cultivation medium was supplemented with 2% Celite® Hyflo Supercel. Separation into culture filtrate and mycelium by multiple sheet filtration (Pall T 1500 filter plates; relative retention range 10–30 μm) yielded 14 L of culture filtrate. The pH of the filtrate was adjusted to pH 5 with 1 M HCl and extracted five times for 30 min with each 2.0 L of ethyl acetate. The organic phases were combined and concentrated under reduced pressure to yield 3.8 g of extract. 1 g extract dissolved in 2 mL of methanol was loaded onto a Sephadex LH-20 gravity column (5 cm × 95 cm, mobile phase methanol, flow rate 102 mL/h, 4 fractions per h). Bioactive fractions were combined to yield 20 mg of a highly purified fraction. This fraction was further refined using semi-preparative HPLC (injection valve Valco C6UW with 5 mL sample loop; Chromatograph LaPrep with two P119 pumps (VWR, Darmstadt), a dynamic mixing chamber (Knauer, Berlin), P314 2-channel-UV-Vis detector (VWR; detection at 230 and 280 nm) with 2-channel-recorder (N-2, Abimed, Langenfeld) on a Luna C-18 (phenomenex; 5 μm, 100 Å; 10 mm x 250 mm) using isocratic separation at 40% acetonitrile in water at

9.45 mL/min for 10 min (column oven 30˚C). The active compound eluted at $t_R$ 7.2 min. Fractions were collected peak based by hand, resulting in 8 mg of pure oxazolomycin A.

## Analysis and structure elucidation

HPLC-HR-MS analysis. Samples were injected into a Dionex Ultimate 3000 HPLC system (Thermo Fisher Scientific), coupled to a maXis 4G ESI-QTOF mass spectrometer (Bruker Daltonics). ESI (pos. and neg. ionization) was performed in Ultra Scan mode with capillary voltage 3.5 kV, nebulizer pressure 2.0 bar, drying gas 8.0 L/min at 350 ˚C. MS/MS spectra were recorded in auto MS/MS mode with collision energy stepping enabled. Molecular formulae were calculated from monoisotopic masses using the SmartFormula function of DataAnalysis (Bruker Daltonics). For analysis of agar extracts of strain Tü2700, a 10 $mm^2$ agar block of Tü2700 colony growing on ISP2 agar for 8 days was extracted with 2 mL of methanol/acetone (1:1) for 1 h in an ultrasonic bath. After drying of the solvent using a rotary evaporator, the extract was reconstituted in 50 μL of methanol and analysed using an Agilent 1200 series HPLC coupled to an Agilent 6330 ion trap (alternating pos. and neg. mode ultra scan, cap. voltage 3.5 kV, temp. 350˚C). NMR analysis. NMR experiments were performed on a Bruker Avance III HD 400 spectrometer at 400 MHz ($^1$H). The compounds were dissolved in methanol-$d_4$. The solution was filtered through cotton wool and transferred into an NMR tube. The chemical shifts were referenced to the residual protonated solvent signals of methanol-$d_4$ ($\delta_H$ = 3.31 ppm and 4.78 ppm; $\delta_C$ = 49.15 ppm).

# Results and discussion

## Development of a cultivation and screening system

**Formatting of the actinomycetes strain collection.**   1232 strains of the Tübingen actinomycetes strain collection were plated on ISP2 agar, and the time from incubation to sporulation was recorded. As the production of specialized metabolites is growth phase-dependent, it is important to assay the bioactivity of a strain at a defined time point relative to the onset of sporulation [32]. Accordingly, the spore suspensions generated from the strains were sorted depending on their sporulation delay, and compiled into spore suspension master plates containing strains with quick (5 days; 29% of the strains), intermediate (7–8 days; 64%), or slow sporulation (10 days; 7%). As master plates, polystyrene transparent square 96-well half-deepwell microplates with flat bottoms were used, to allow cooling of the master plates during transfer using pre-cooled aluminum blocks. The spore suspensions were filled into the plates (96 spore suspensions per master plate, 1.1 mL per well), which were covered with silicon lids and stored at -80˚C.

**Inoculation of agar plugs and strain cultivation.**   In a first experiment, a standard 96-well MTP was filled with liquefied ISP-2 agar (>85˚C, 160 μL per well). After solidification of the agar at room temperature, a commercially available cryo-replicator was found to be suitable to transfer the spore suspensions from the master plates to the agar-containing plates (Fig 1B and 1C). The replicator consists of 96 individual spring-borne stainless steel pins. For inoculation, the cryo-replicator is pressed (app. 0.2 N) for 3 s onto the frozen surface of the spore stock. This leads to the melting of app. 0.3 μl of the spore stock, which will form an app. 50 μm film on the tip of the pin [39]. These spore suspension films were then used to inoculate the agar plugs in the MTPs. Care needed to be taken not to lower the pins too much to avoid damaging the agar plugs. Before switching to a new spore suspension master plate, the cryo-replicator was heat sterilized and allowed to cool down to ambient temperature. To analyze the possibility of cross-contamination during inoculation, MTPs containing an actinomycete spore suspension stock only in every third well were prepared. This master plate was used for

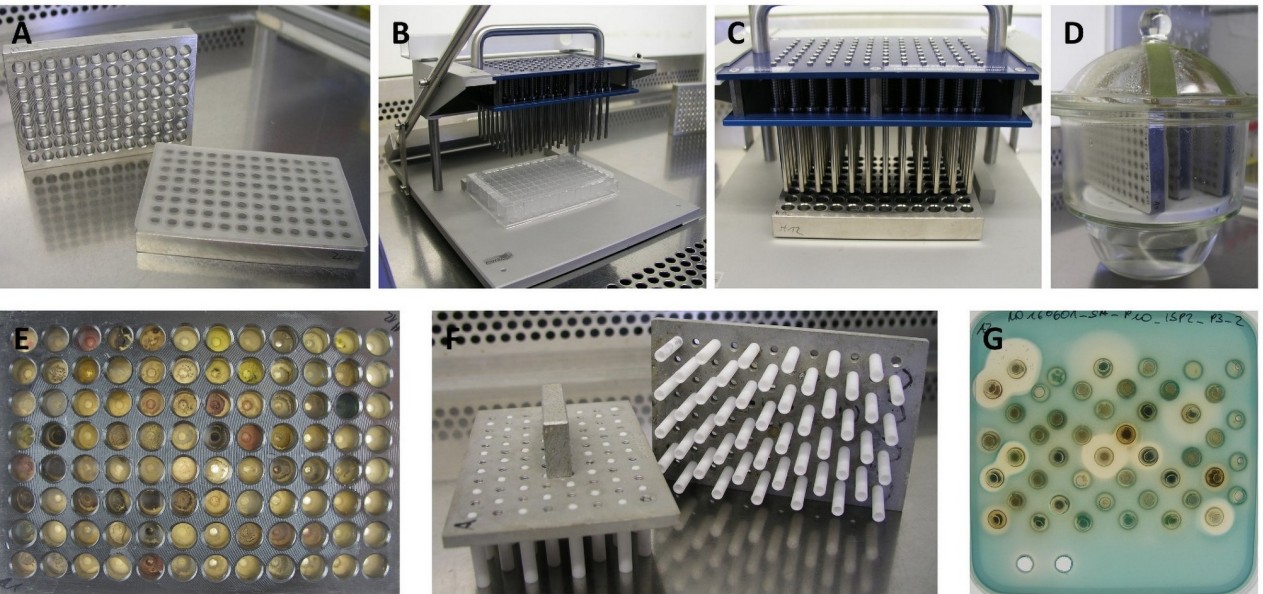

**Fig 1. "From spore to bioactivity"—The workflow of the combined MTP cultivation system and an agar plug diffusion assay.** (A) Bottomless, stainless steel MTPs are closed on one side with silicone mats. The wells are filled with agar. (B) For inoculation, a cryo-replicator is first pressed onto the frozen surfaces of the spore suspensions. (C) Subsequently, the replicator pins are lowered onto the agar surface in the MTPs. (D) The plates are sealed with a gas-permeable membrane and stored in a water-saturated atmosphere. (E) After an appropriate incubation time, the MTPs are unsealed, and the growth of the bacteria is visually evaluated. (F) With the help of two plunger devices, agar plugs are pushed out into two square petri dishes. (G) An agar-plug assay reveals actinomycete strains secreting active natural products into their solid growth medium.

three successive inoculations of agar-filled MTPs. We did not observe any contamination of adjoining wells with viable actinomycetes spores. However, although the growth of the actinomycete strains in the MTP wells was satisfactory, we found it was not possible to remove the agar plugs from the wells and put them into petri dishes without destroying the plugs.

Removal of the bottoms of the wells was the solution to this problem, as it enabled us to push out the agar plugs. As bottomless MTPs are not commercially available, we removed the bottoms of standard polystyrene MTPs manually using a drill with the same diameter as the wells. It was not possible to remove the bottoms in an aseptic environment, thus a subsequent sterilization step was needed. Polystyrene is not heat stable, thus chemical sterilization according to the Centers of Disease Control and Prevention (CDC) Guidelines, employing 7.5% $H_2O_2$ for 15 min and subsequently 70% ethanol for 5 min to rinse away the $H_2O_2$, was used [49]. However, subsequent inoculation experiments with these plates showed that this sterilization protocol was not sufficient, and frequent contamination especially with fungi was observed. We hypothesize that small scratches in the polystyrene, originating from the drilling process, cannot be reliably sterilized with the liquid disinfectants. After some trial and error, we designed bottom-less stainless-steel MTPs (96-well, round). These plates were reusable and could be heat sterilized at 160°C for 2 h. No contaminations were observed after we introduced these stainless-steel plates into our workflow (Fig 1A). Before filling the agar into the plates, the bottoms of the wells were closed with a silicone mat. Each well was filled with 160 μl of agar. Any medium can be used, resulting in a high flexibility of the growth system. Increasing the amount of agar in the medium from the commonly used 2.0% to 2.5% resulted in agar plugs with improved stability during subsequent manipulations.

The inoculated MTPs were closed with gas-permeable membranes (heat sealed rayon fibers) coated with acrylate adhesive, and incubated at 27˚C for 5 to 10 days, depending on the sporulation delay. To prevent the agar from drying out during incubation, the MTPs were incubated in a water-saturated atmosphere (Fig 1D).

**Transfer of plugs to the agar-plug assay.** After incubation, the agar plugs were transferred from the growth plate into a petri dish. To achieve this, both sealing mats were removed. After a visual examination to confirm satisfactory growth (Fig 1E), the agar plugs were pressed out of the plate, directly into a petri dish, using a plunger device. A 10 cm square petri dish has a suitable width to accommodate the agar plugs from an MTP. We noticed that if all 96 agar plugs are extruded at the same time, they are too close together to allow the observation of distinct halos around the agar plugs in the subsequent bioactivity assay. Thus, we designed the plunger device in a way that only every other agar plug is pushed out at a time. Therefore, two plunger devices with 48 plungers each were needed (A1 to H11, and A2 to H12), and one MTP results in two petri dishes for bioactivity testing. The distances of the agar plugs in the petri dish were well suited for a subsequent agar-plug assay.

We found that the agar plugs were sticking strongly to the plunger tips of our first prototype, making it very difficult to reliably place the plugs in the petri dish. We solved this problem by making the tips of the plungers concave, so that only the outer rim of the plunger tip comes into direct contact with the agar plug (Fig 1F). Reducing the contact area this way, the adhesion of the bottom of the agar plug to the petri dish surface became strong enough, so that the agar plugs stayed in place after removal of the plunger device. The device was constructed to be heat sterilizable (aluminum handle, Teflon plungers).

Liquefied agar (42˚C, 40 mL) containing a monitor strain was carefully poured into the petri dish around the agar plugs, and kept at room temperature for 1 h to allow solidification of the agar. Subsequently, the petri dish was incubated at the conditions required by the monitor strain. This workflow allowed the screening of the actinomycete strain collection for any bioactivity of interest that can be detected in agar-plus assays, as described in the following.

## Screening of an actinomycete strain collection for QS inhibitors

To demonstrate the usability of the developed workflow, we performed a bioactivity screening campaign using *S. aureus* PC322 as monitor strain [46]. *S. aureus* PC322 can be used to examine the effects of compounds on virulence gene expression in *S. aureus*, as it is a transcriptional reporter strain carrying *lacZ* (β-galactosidase) fused to the central virulence gene *hla* (α-hemolysin). Transcription of *hla* is regulated via the Agr (auto-inducing peptide, AIP) QS pathway [50], thus compounds inhibiting staphylococcal AIP-based QS lead to reduced *hla* expression. Lower *hla* expression, in turn, leads to reduced *lacZ* expression in the *hla*::*lacZ* monitor strain and subsequently reduced degradation of the chromogenic β-galactosidase substrate X-Gal. The assay has two potential read-outs: If there is a clear and colorless inhibition zone around a test compound, the compound is antibacterial against *S. aureus*. If the halo is colorless but hazy, the compound can be suspected to have inhibitory activity on Agr-based QS without being directly antibacterial [46].

**Screening of the strain collection.** It is known that the medium composition has a strong influence on the natural product spectrum an actinomycete strain produces. Thus, we decided to screen the 1232 strains of the Tübingen actinomycetes strain collection we had formatted into spore suspension master plates after growth on two different solid media (ISP2 and ISP3). The cultivation and the screening assays were performed in triplicate. Strains that were reproducibly active were counted as "hit strains". Independent from the cultivation medium, 82 strains (7%) were found to inhibit QS (hazy halo), 284 strains (23%) displayed antibiotic

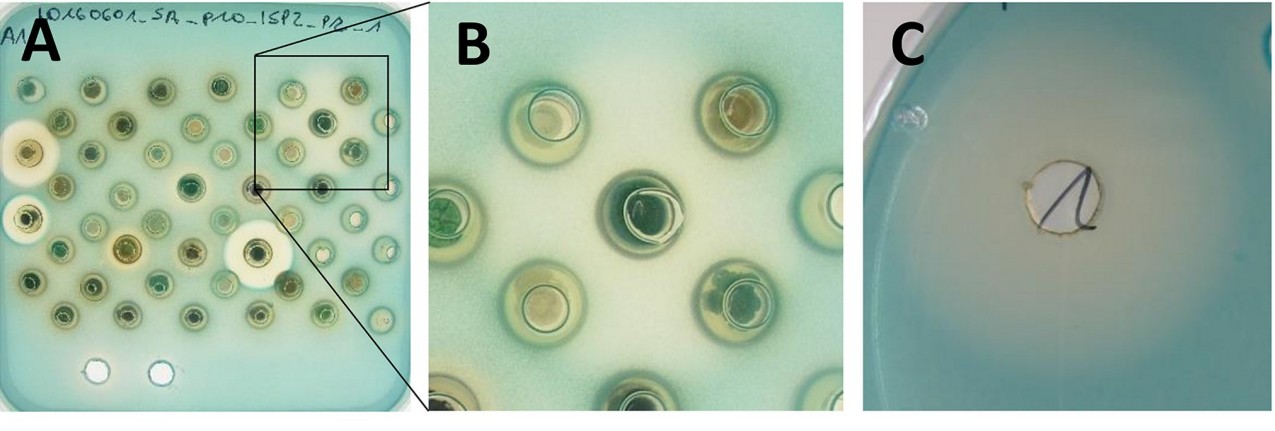

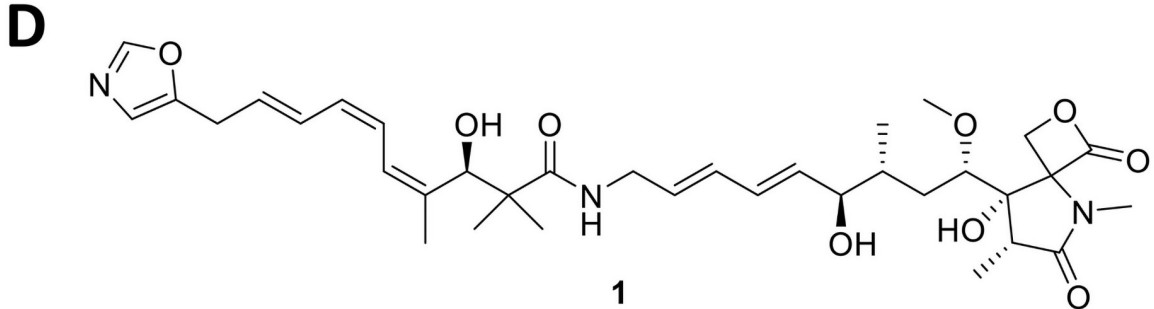

**Fig 2.** (A) Screening plate containing the agar plug of Tü2700 (hazy halo on the top right; note the differences to the clear halos of three other strains indicating direct antibacterial activity). (B) Detailed view of the inhibition zone of Tü2700. (C) The same hazy halo (inhibitory activity of the strain on *LacZ* expression) was also observed when the strain was grown submersed. (D) Structure of Oxazolomycin A (**1**).

activity (clear inhibition zone), and 866 strains (70%) did not show any activity in the assay. The high proportion of antibacterial strains was not surprising, as the Tübingen actinomycete strain collection contains many strains that were selected for their antibiotic activity.

A majority of the strains with QS inhibitory activity (60 strains, 73%) exhibited only modest halos (< 5 mm), 17 strains (21%) showed halos > 10 mm, and 5 strains (6%) showed halos > 20 mm. While three of the most active strains (Tü1792, Tü2698, and Tü2700) showed activity when cultivated on either agar, Tü3337 was only active when grown on ISP2 agar, and Tü 99 only when grown on ISP3 agar. According to 16S rRNA analysis, all of the 5 most active strains belong to the genus *Streptomyces* (S1 Fig in S1 File). With the largest halo of the tested strains (25 mm, Fig 2A and 2B), *Streptomyces* Tü2700 was chosen for identification of the active compound(s).

**Taxonomic characterization of Tü2700.** Whole-genome sequencing and phylogeny analysis using autoMLST showed the strain to belong to the genus *Streptomyces*, the closest type strains being *S. alboflavus* NRRL B-2373 (average nucleotide identity (ANI) of 86.0%, mash distance 0.1404) and *S. aureocirculatus* NRRL ISP5386 (ANI 85.3%, mash distance 0.1475) (see S1 Fig in S1 File). These results indicate that *Streptomyces* Tü2700 apparently belongs to a yet undescribed species. However, to confirm this hypothesis, further experiments to determine e.g. its lipid profile, morphology, and its utilization of different carbon sources would be needed, which was not within the scope of this study.

**Cultivation of Tü2700 and isolation of the active compound.** Isolation of a bioactive compound from liquid cultivation medium is more convenient compared to solid medium.

Thus, we tested whether the active compound of *Streptomyces* Tü2700 is also produced in liquid medium. Indeed, we found that the same hazy halo could be observed when the strain was grown submersed (Fig 2C). To assess the influence of different cultivation media on the production of the putatively QS inhibiting compound by *Streptomyces* Tü2700, the strain was cultivated in various liquid media (NL19, NL200, NL300, NL333, NL400, NL410, NL500, ISP3). NL19 was found to be a suitable medium for submersed cultivation of Tü2700 and production of the active compound over a cultivation time of 4 days. Bioactivity-guided isolation resulted in the identification of oxazolomycin A (**1**, structure see Fig 2D) as the main active compound produced by this strain. Subsequent analysis of a *Streptomyces* Tü2700 agar plug by HPLC-MS confirmed that **1** is also produced when the strain grows on solid medium (see SI).

The identity of **1** was confirmed based on UV spectroscopy, mass spectrometry, and NMR spectroscopy data (S2–S4 Figs in S1 File), which are in agreement with previously reported data of oxazolomycin A [51, 52]. Oxazolomycin A is a well-known compound from *Streptomyces*, first isolated in 1985 [53]. It is known to have antibiotic, antiviral, and antitumor activity [52]. These activities are most likely caused by the property of **1** to act as a protonophore at pH $< 7.0$ and conveys both protons and monovalent cations such as potassium at pH $> 7.5$ [54]. Thus, the activity we observed for sub-lethal concentrations of 1 (50 µg/mL) on *S. aureus* PC322 is most likely also due to this unspecific protonophore/ionophore activity rather than specific inhibition of the QS of *S. aurus*, demonstrating the limitation of the monitor strain used in this screening against unspecific growth inhibitors which give false-positive results.

## Conclusions

Conventional cultivation and screening systems for actinomycetes use submersed cultivation of the test strains. While actinomycetes can grow and produce secondary metabolites in submersed cultures, they naturally grow attached to solid surfaces. It is known that cultivation conditions have a tremendous influence on the production of secondary metabolites. To the best of our knowledge, there has not yet been a convenient way to combine agar diffusion assays with microtiter-plate-based cultivation systems in which strains can grow on a solid surface. Thus, we developed a cultivation system for actinomycetes which allows the cultivation of 96 different strains on solid agar in stainless steel microtiter plates, complementing the existing cultivation systems. The developed cultivation system has been seamlessly integrated with an agar-plug assay screening, resulting in a workflow that can conveniently be used to screen natural product-producing microorganisms in medium throughput.

The developed cultivation and screening system was tested on the Tübingen actinomycete strain collection. We found that the system can conveniently be used to cultivate $> 1000$ strains in parallel on different agars, and to directly use the resulting inoculates in agar-plug assays. As proof of concept, we used an agar-plug assay with the monitor strain *S. aureus* PC322, which has been developed to identify compounds inhibiting Agr based quorum sensing.

While the screening presented here did not result in the discovery of a novel compound or a compound with specific activity, the successful screening campaign showed the general value of the developed screening system. It allowed rapid and comprehensive initial screening of the strain collection against a monitor strain in an agar-plus assay. Since the development of the system, we have conducted 3 screening campaigns. The results of these campaigns will be reported elsewhere or have already been published [55]. Sterilization of the replicator proved to be the rate-limiting step in the developed workflow, but inoculating multiple plates from the same pore suspension master plate is a rapid process. Thus, increasing the number of media studied in parallel or parallelizing the number of different monitor strains tested against can increase the screening speed as needed.

## Supporting information

**S1 File. The phylogenetic tree of Streptomyces Tü2700 as well as UV/MS/NMR spectra of 1 and results of an HPLC analyses of a Tü2700 agar plug can be found in the Supporting Information.**
(PDF)

## Acknowledgments

We thank T. Mourlidis for manufacturing the stainless steel microtiter plates and the plunger devices.

## Author Contributions

**Conceptualization:** Nico Ortlieb, Timo Horst Johannes Niedermeyer.

**Data curation:** Nico Ortlieb, Elke Klenk.

**Formal analysis:** Nico Ortlieb, Andreas Kulik, Timo Horst Johannes Niedermeyer.

**Funding acquisition:** Timo Horst Johannes Niedermeyer.

**Investigation:** Nico Ortlieb, Andreas Kulik.

**Methodology:** Nico Ortlieb, Elke Klenk, Timo Horst Johannes Niedermeyer.

**Project administration:** Timo Horst Johannes Niedermeyer.

**Resources:** Timo Horst Johannes Niedermeyer.

**Supervision:** Timo Horst Johannes Niedermeyer.

**Visualization:** Nico Ortlieb, Timo Horst Johannes Niedermeyer.

**Writing – original draft:** Timo Horst Johannes Niedermeyer.

**Writing – review & editing:** Nico Ortlieb, Timo Horst Johannes Niedermeyer.

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
