## [Decision Letter · Decision Letter 0]

27 Aug 2021

PONE-D-21-25646

Development of an agar-plug cultivation system for bioactivity assays of actinomycete strain collections

PLOS ONE

Dear Dr. Niedermeyer,

Thank you for submitting your manuscript to PLOS ONE. After careful consideration, we feel that it has merit but does not fully meet PLOS ONE’s publication criteria as it currently stands. Therefore, we invite you to submit a revised version of the manuscript that addresses the points raised during the review process.

The reviews received indicate that minor revisions are needed.

We look forward to receiving your revised manuscript.

Kind regards,

Marcos Pileggi, Ph.D

Academic Editor

PLOS ONE

Journal Requirements:

2. Please ensure that you refer to Figure 2 in your text as, if accepted, production will need this reference to link the reader to the figure.

3. We note that Figure 1 in your submission contain copyrighted images. All PLOS content is published under the Creative Commons Attribution License (CC BY 4.0), which means that the manuscript, images, and Supporting Information files will be freely available online, and any third party is permitted to access, download, copy, distribute, and use these materials in any way, even commercially, with proper attribution. For more information, see our copyright guidelines: http://journals.plos.org/plosone/s/licenses-and-copyright.

Reviewers' comments:

Reviewer's Responses to Questions

Comments to the Author

1. Is the manuscript technically sound, and do the data support the conclusions?

Reviewer #1: Yes

Reviewer #2: Partly

2. Has the statistical analysis been performed appropriately and rigorously?

Reviewer #1: N/A

Reviewer #2: N/A

3. Have the authors made all data underlying the findings in their manuscript fully available?

Reviewer #1: Yes

Reviewer #2: Yes

4. Is the manuscript presented in an intelligible fashion and written in standard English?

Reviewer #1: Yes

Reviewer #2: Yes

5. Review Comments to the Author

Reviewer #1: The authors report the development of a medium-throughput cultivation and agar-plug assay workflow, which enables the convenient inoculation of agar plugs with actinomycete spore suspensions from a strain collection and the convenient transfer of agar plugs to petri dishes to carry out agar-plug mediated bioactivity assays. To demonstrate the usability of the developed workflow, they performed a bioactivity screening campaign of 1232 collected actinomycete strains using Staphylococcus aureus PC322 as a monitor strain. The results showed that 30% of the tested strains were found to display antibiotic activity or inhibit Agr-based quorum sensing (QS). Moreover, they showed that all of the 5 most active strains belong to the genus Streptomyces based on 16S rRNA analysis. Finally, the genome of strain Streptomyces Tü2700 was sequenced and phylogeny analysis was performed and then the main active compound produced by this strain was identified. Overall, this study is well-done and the data is convincing, which provides a novel workflow to screen natural-product microbes in medium throughput and allows the convenient initial screening of strain collections in an agar plug bioactivity assay. I recommend its publication, however, before that, the following comments should be addressed.

1. The abstract part should be optimized. The results of this study should be described. In addition, the authors should also describe briefly what measures the authors take to address the challenges encountered in the workflow.

2. Line 85-86, the sentence “a plug---out of the agar” should be rephrased.

3. Line 130, the strain name S. aureus should be provided in full name for the first time.

4. Line 135, OD what is the wavelength?

5. The result part “Screening of the strain collection”, how many strains were used here? The number should be provided.

Reviewer #2: Dear Authors,

The paper entitled “Development of an agar-plug cultivation system for bioactivity assays of actinomycete strain collections” describes the development and application of a medium-throughput agar-plug based screening system for the identification of bioactive compounds from actinomycete strains cultured on solid media. Using a collection of bespoke devices optimised for repeatability and sterilisation the authors were able to identify strains exhibiting quorum sensing inhibitory activity using a S. aureus hla::lacZ reporter strain. One such strain, Streptomyces sp. Tü2700, was shown to produce strong bioactivity which, following further isolation, the authors identified as oxazolomycin A.

This submission recognises its limitations as primarily a methodology paper and whilst short covers the necessary material. I believe papers like this are important contributions to the research community and this submission is no different. In the current state I would recommend that this paper is accepted for submission following minor revisions.

In my opinion the greatest outstanding issue of this paper is in the identification of oxazolomycin A as the bioactive compound. Whilst oxazolomycin A is clearly the compound they have managed to isolate from Tü2700 this is following a series of liquid culture optimisations for isolation and structural elucidation. The link to it being the bioactive compound identified in the screen is unsubstantiated. This doesn’t actually matter per se however I would recommend the authors make this distinction clearer and that the screen only allows for the identification of strains with the propensity to produce bioactive compounds but what is subsequently identified in the liquid isolation is most likely different to that showing activity on the agar plug screen. If the authors wished to confirm Tü2700 was producing oxazolomycin A during the screen it would be prudent to isolate the compound directly from the separate agar plug plates which, whilst it may take a significant number of plates, has been successfully achieved for years.

In addition to this I present below a few minor points for consideration:

- The authors make the claims (e.g. line 25, 72) that the solid cultivation of actinomycetes is rarely used for bioactivity screenings. Whilst this may be true for extremely large scale screens I would argue that the classical overlay bioassay is a widespread and is indeed first line methodology for screening the bioactivity of new isolates due to its simplicity and replicability. I would recommend the authors to readjust their claims slightly.

- I believe the manuscript would benefit from a small explanation of inhibitors of bacterial quorum sensing, perhaps just a few sentences, to give the readers a better understanding of why they are a promising source of therapeutics.

- I believe a Figure containing images of the strain Tü2700 and its bioactivity as noted from the screen would be of great benefit to the readers.

- Some small grammar issues:

o Line 25: I think it would be more correct to use “submersed” rather than “submerse” and this is continued throughout the manuscript

o Line 107: Please be consistent in the use of ml or mL

o Line 151: Similarly with v/v or V/V

I thank the authors for their submission, I believe the methodology, and general results, presented here may be of great benefit to the actinomycete research community.

6. PLOS authors have the option to publish the peer review history of their article (what does this mean?). If published, this will include your full peer review and any attached files.

Do you want your identity to be public for this peer review? For information about this choice, including consent withdrawal, please see our Privacy Policy.

Reviewer #1: No

Reviewer #2: Yes: Thomas C McLean

---

## [Decision Letter · Decision Letter 1]

11 Oct 2021

Development of an agar-plug cultivation system for bioactivity assays of actinomycete strain collections

PONE-D-21-25646R1

Dear Dr. Niedermeyer,

We’re pleased to inform you that your manuscript has been judged scientifically suitable for publication and will be formally accepted for publication once it meets all outstanding technical requirements.

Kind regards,

Marcos Pileggi, Ph.D

Academic Editor

PLOS ONE

Additional Editor Comments (optional):

Reviewers' comments:

Reviewer's Responses to Questions

**Comments to the Author**

1. If the authors have adequately addressed your comments raised in a previous round of review and you feel that this manuscript is now acceptable for publication, you may indicate that here to bypass the “Comments to the Author” section, enter your conflict of interest statement in the “Confidential to Editor” section, and submit your "Accept" recommendation.

Reviewer #1: All comments have been addressed

Reviewer #2: All comments have been addressed

2. Is the manuscript technically sound, and do the data support the conclusions?

Reviewer #1: Yes

Reviewer #2: Yes

3. Has the statistical analysis been performed appropriately and rigorously? 

Reviewer #1: Yes

Reviewer #2: N/A

4. Have the authors made all data underlying the findings in their manuscript fully available?

Reviewer #1: Yes

Reviewer #2: Yes

5. Is the manuscript presented in an intelligible fashion and written in standard English?

Reviewer #1: Yes

Reviewer #2: Yes

6. Review Comments to the Author

Reviewer #1: (No Response)

Reviewer #2: Deat Authors,

Thank you for your revisions following review. I am very happy with the state of the manuscript and suggest just a few minor grammatical modifications to further improve your manuscript:

Line 56 - "sometimes comprising - of - tens of thousands of strains"

Line 85 - Actinobacteria should be capitilised

Line 107 - "We describe the successful development - of - a medium"

Line 137 - Remove the full stop after Staphylococcus

Line 152 - Please format H2O correctly

Lines 160,161,167,182 - Please capitilise L for litre

Line 282 - I suggest you change this sentance to read "Transcription of hla is regulated..." this avoid capitilising a gene name

Line 320 - Change submers to submersed

Line 345 - Change "are using" to use

With that in effect I would like to thank the authors for their swift and precise work and commend them on the final manuscript, following my colleague reviewers' comments.

7. PLOS authors have the option to publish the peer review history of their article (what does this mean?). If published, this will include your full peer review and any attached files.

Reviewer #1: No

Reviewer #2: **Yes: **Thomas C. McLean

---

## [Editor Report · Acceptance letter]

18 Oct 2021

PONE-D-21-25646R1 

Development of an agar-plug cultivation system for bioactivity assays of actinomycete strain collections 

Dear Dr. Niedermeyer:

I'm pleased to inform you that your manuscript has been deemed suitable for publication in PLOS ONE. Congratulations! Your manuscript is now with our production department. 

Kind regards, 

on behalf of

Dr. Marcos Pileggi 

Academic Editor

PLOS ONE